Differential expression of miR-1, a putative tumor suppressing microRNA, in cancer resistant and cancer susceptible mice

Fleming Jessica L. 1
Gable Dustin L. 2
Samadzadeh-Tarighat Somayeh 3
Cheng Luke 2
Yu Lianbo 4
Gillespie Jessica L. 1
Toland Amanda Ewart 1 5 Amanda.toland@osumc.edu
1 Department of Molecular Virology, Immunology and Medical Genetics, The Ohio State University Comprehensive Cancer Center, The Ohio State University , Columbus, OH , USA
2 Biomedical Science Program, The Ohio State University , Columbus, OH , USA
3 Division of Hematology/Oncology, Department of Internal Medicine, The Ohio State University , Columbus, OH , USA
4 The Center for Biostatistics, The Ohio State University , Columbus, OH , USA
5 Division of Human Genetics, Department of Internal Medicine, The Ohio State University , Columbus, OH , USA
Hoheisel Jörg
Electronic publication date: 2013 Apr 16
Publication date: 2013
Volume: 1
Electronic Location ID: e68
Received 2012 Oct 12; Accepted 2013 Mar 25
Copyright: © 2013 Fleming et al.
Copyright year: 2013
Copyright holder: Fleming et al.
License: This is an open access article distributed under the terms of the Creative Commons Attribution License, which permits unrestricted use, distribution, and reproduction in any medium, provided the original author and source are credited.
License URL: https://creativecommons.org/licenses/by/3.0/

Keywords: microRNA, Skin cancer, miR-1

Funding: American Cancer Society RSG-07-083-01-MGO National Institutes of Health CA134461 OSU Comprehensive Cancer Center Pelotonia Fellowship Up on the Roof Fellowship This work was supported in part by the American Cancer Society [RSG-07-083-01-MGO], the National Institutes of Health [CA134461] and the OSU Comprehensive Cancer Center. DG was supported by a Pelotonia Fellowship, and JF by an Up on the Roof Fellowship. The funders had no role in study design, data collection and analysis, decision to publish, or preparation of the manuscript.

==============================
Mus spretus mice are highly resistant to several types of cancer compared to Mus musculus mice. To determine whether differences in microRNA (miRNA) expression account for some of the differences in observed skin cancer susceptibility between the strains, we performed miRNA expression profiling of skin RNA for over 300 miRNAs. Five miRNAs, miR-1, miR-124a-3, miR-133a, miR-134, miR-206, were differentially expressed by array and/or qPCR. miR-1 was previously shown to have tumor suppressing abilities in multiple tumor types. We found miR-1 expression to be lower in mouse cutaneous squamous cell carcinomas (cSCCs) compared to normal skin. Based on the literature and our expression data, we performed detailed studies on predicted miR-1 targets and evaluated the effect of miR-1 expression on two murine cSCC cell lines, A5 and B9. Following transfection of miR-1, we found decreased mRNA expression of three validated miR-1 targets, Met, Twf1 and Ets1 and one novel target Bag4. Decreased expression of Ets1 was confirmed by Western analysis and by 3’ reporter luciferase assays containing wildtype and mutated Ets1 3’UTR. We evaluated the effect of miR-1 on multiple tumor phenotypes including apoptosis, proliferation, cell cycle and migration. In A5 cells, expression of miR-1 led to decreased proliferation compared to a control miR. miR-1 expression also led to increased apoptosis at later time points (72 and 96 h) and to a decrease in cells in S-phase. In summary, we identified five miRNAs with differential expression between cancer resistant and cancer susceptible mice and found that miR-1, a candidate tumor suppressor, has targets with defined roles in tumorigenesis.

Introduction

Mus Spretus mice are resistant to several cancer types including skin, colon, lung and thymic lymphoma (Nagase et al., 1995; Manenti et al., 1996; Villa-Morales, Santos & Fernandez-Piqueras, 2006; Huang et al., 2007). Due to its cancer resistant nature M. Spretus has been used for quantitative trait locus (QTL) mapping for cancer susceptibility loci (Dejager, Libert & Montagutelli, 2009). To date, most of the QTL studies for cancer susceptibility have focused on identification of candidate genes with amino acid substitutions or differences in mRNA expression. Recent genome-wide association studies for cancer risk in humans have identified cancer-associated polymorphisms with roles in regulation of gene expression (Pomerantz et al., 2009; Spain et al., 2012).

MicroRNAs (miRNAs) are short RNAs of 20-22 nucleotides with well-documented roles in gene regulation (Siomi & Siomi, 2010). They bind to the 3’untranslated region (3’UTR) of genes and may be involved in binding to other parts of the mRNA as well (Wery, Kwapisz & Morillon, 2011). Binding results in mRNA degradation or impaired translation and subsequent decreased protein expression (Shyu, Wilkinson & van Hoof, 2008; Djuranovic, Nahvi & Green, 2011). Increasing evidence indicates that miRNAs play key roles in carcinogenesis. Expression profiling studies have demonstrated that many miRNAs are down-regulated during tumor development, resulting in subsequent up-regulation of their target genes and respective proteins. These miRNAs act as tumor suppressors and target cell cycle, apoptosis, proliferation, invasion and metastasis genes (Croce, 2009). Likewise, another subset of miRNAs is up-regulated during tumorigenesis resulting in down-regulation of their targets which are frequently tumor suppressor genes (Medina & Slack, 2008). Previous studies indicate that several miRNAs map in close proximity to mouse QTLs for cancer susceptibility suggesting that variations in miRNA sequence or expression may be important for cancer susceptibility (Sevignani et al., 2007).

Because of the strong correlation between miRNA expression and tumorigenesis, we hypothesized that miRNAs showing differential expression between skin cancer resistant M. spretus (SPRET/EiJ) mice and skin cancer susceptible M. musculus (FVB/NJ) mice could be considered as candidates for cancer susceptibility. To identify miRNAs which may play a role in differences in cancer susceptibility between SPRET/EiJ and FVB/NJ, we performed expression profiling on normal skin samples from these strains of mice. Five differentially expressed miRNAs with described roles in tumorigenesis were identified and validated. Based on our observations and reports in the literature supporting miR-1 as having tumor suppressor function in a variety of cancer types (Nasser et al., 2008; Datta et al., 2008; Yan et al., 2009; Li et al., 2012; Hudson et al., 2012; Nohata et al., 2011; Nohata et al., 2012a; Nohata et al., 2012b; Tominaga et al., 2013), we hypothesized that miR-1 acts as a tumor suppressor in cutaneous squamous cell carcinoma (cSCC). Here, we describe the effects of expressing miR-1 in cSCC cells and the identification of Ets1, as a miR-1 target in the mouse.

Materials and Methods

RNA isolation

Animal studies were approved by the Ohio State University (OSU) and University of California, San Francisco Institutional Animal Care and Use Committees. The OSU IACUC determined that the research performed at OSU was exempt from IACUC review as pre-existing or commercially available animal specimens were used for this study. Fresh frozen skin samples from three FVB/NJ and three SPRET/EiJ female mice aged 4–5 weeks were obtained through the Jackson Laboratory surgical service. RNA was isolated from the skin using standard Trizol extractions (Invitrogen, Grand Island, NY). We isolated RNA a second time from two of the skin samples to generate a replicate control for the microarrays. Samples were DNAase treated. RNA was quantitated using Nanodrop (ThermoScientific, Wilmington, DE). RNA was isolated from eleven cSCCs from chemically treated C57Bl6/FVB F1 mice using standard Trizol methods. RNA samples were DNAase treated.

MicroRNA expression arrays and data analysis

Five µg of total RNA from each sample was sent to the OSU Comprehensive Cancer Center MicroArray Shared Resource for miRNA expression analysis using the miRNA microarray chip OSUCCC version 4.0 (Liu et al., 2004; Liu et al., 2008). For each strain of mice, a fourth RNA sample of a second independent RNA isolation was also evaluated. The arrays contain over 300 human and mouse miRNAs spotted in triplicate with probes for both precursor and mature miRNAs. Experiments were performed and analyzed as previously described (Liu et al., 2004; Liu et al., 2008). In brief, a linear model was employed to detect differentially expressed miRNAs. In order to improve the estimates of variability and statistical tests for differential expression, a variance smoothing method with fully moderated t-statistic was employed for this study (Yu et al., 2011). The significance level was adjusted by controlling the mean number of false positives (Gordon et al., 2007).

Quantitative PCR

Quantitative PCR (qPCR) validation of candidate miRNAs. qPCR was performed using Applied Biosystems TaqMan probes to validate expression of the six candidate miRNAs (Foster City, CA). Probes were specific to either mature (miR-1, miR-133a, miR-124a-3, miR-206) or precursor (miR-134, miR-206, miR-9-1) miRNAs concordant with the array results. Reverse transcription for mature miRNAs was performed using Applied Biosystem’s High Capacity cDNA Reverse Transcription reagents according to manufacturer’s protocol in 7.5 µl reactions (Carlsbad, CA). Three independent skin RNA samples were used for each strain from samples not used on the original arrays. qPCR of miR-1 levels from eleven primary mouse cSCCs was conducted as for normal skin. For miRNA quantification, real-time PCR was performed using Applied Biosystem TaqMan Assays per manufacturer’s recommended conditions. All samples were done in triplicate. qPCR runs included no template and no-reverse transcriptase controls. Expression was normalized to expression levels for sno202 RNA for mature miRNAs or to L19 for pri-miRNAs. qPCR was conducted using Applied Biosystems 7900HT instrument. Cycle threshold (CT) values were averaged across triplicates and delta CT values were calculated between each test miRNA and control. The fold expression of SPRET/EiJ miRNA relative to FVB/NJ was calculated. Standard deviations of fold differences of the percentage of reference gene expression were made by comparisons between independent RNA samples of each of the test samples. Student’s t-test was used to calculate p-values for the qPCR. A nonparametric Mann-Whitney U test was performed to test for differences between miR-1 expression in tumors versus SPRET/EiJ and FVB normal skin.

qPCR of predicted mRNA targets. To measure expression of predicted targets, mRNA expression was measured at 48 and 72 h post miR-1 or scrambled precursor miR negative control transfection in A5 cells and at 48, 72 and 96 h post-transfection in B9 cells using SYBR green quantitative real-time PCR. Reverse transcription of mRNA was performed using a Bio-Rad iScript cDNA synthesis kit according to manufacturer’s recommended conditions. Primers were designed using Integrated DNA Technologies. Target gene expression was measured using Bio-Rad SYBR Green Supermix in 10 µl reactions according to manufacturer’s protocol. qPCR of each sample was performed in triplicate and was used to measure Ets1, Met, Bag4, Twf1, Sp1, Taok1, Zfp148 and Trp53 target gene expression. The following primer sets were used. Ets1: Forward: 5’ TGTATGAGTGGAGCAGCACTGTGT 3’, Reverse: 5’ AGGTAGGGTCTCCATTAACCT 3’, Met: Forward: 5’ AACGGGTATTGGGAAGACCCTGAA 3’, Reverse: 5’ ATCCCGTCTAACAGGAAGAAGGCT 3’, Bag4: Forward: 5’ ACTCCACGGAAGTTCCAA ACACCT 3’, Reverse: 5’ TTCCAGGGTTCTGTGAAGCAGGAT 3’, Twf1: Forward 5’ TTCCAGGCTTTGGAGAAGGTGAGT 3’, Reverse: 5’ AGTCTCCTTCGTGGGAATGCTTGT 3’, Sp1: Forward 5’ TGCGGCAAAGTATATGGCAAGACC 3’, Reverse 5’ ACTCCTCATGAAACGCTTAGGGCA 3’, Taok1: Forward 5’ GAACCAGGCCAAGTGAAACTTGCT 3’, Reverse 5’ AAAGGAGGCTTCCTCTCGGCTAAT 3’, Zfp148: Forward 5’ GCACTGCAATGCTGCCTTTAGAAC 3’, Reverse 5’ TTTCGCCGGTATGGATCTTCTCGT 3’, Trp53: Forward 5’ ACAAGAAGTCACAGCACATGACGG 3’, Reverse 5’ TTCCTTCCACCCGGATAAGATGCT 3’, and L19: Forward: 5’ ATTCCCGGGCTCGTTGCCGGAAAA 3’, Reverse: 5’ ATTGGCAGTACCCTTCCTCTTCCCTA 3’. Candidate targets were normalized to L19 expression and fold difference relative to the scrambled control at 48 h post-transfection were calculated. Standard deviations of fold differences of the percentage of reference gene expression were made by comparisons between independent RNA samples. Experiments included no template and no reverse transcriptase controls for each gene. Student’s t-test was used to calculate p-values.

qPCR confirmation of transfections. RNA was isolated from a duplicate well for each mock, scrambled control or miR-1 transfected experiment to confirm increased miR-1 expression in the miR-1 transfected cells. qPCR for miR-1 expression was conducted as described above.

Identification of candidate target genes for differentially expressed miRNAs

Target prediction programs, www.microRNA.org (Beta et al., 2008) and www.targetscan.org were searched for predicted mRNA targets for miR-1 in both mouse and human. Targets for miR-1 were prioritized for further study by the number of predicted binding sites per 3’UTR (>1 site), strength of binding score, and predicted binding of the miRNA in both human and mouse. A literature scan was also performed to identify targets which had previously been validated experimentally.

Cell culture and transient transfections

A5 and B9, mouse cutaneous spindle and cSCC cell lines respectively, and C5N, a non-tumorigenic mouse keratinocyte cell line, were used for these studies (Zoumpourlis et al., 2003). Cell lines were grown in Dulbecco’s medium supplemented with 10% fetal bovine serum and 1% penicillin/streptomycin in an atmosphere of 5% CO2. Transient transfections were performed using Invitrogen Lipofectamine 2000 according to manufacturer’s protocol (Life Technologies, Grand Island, NY). Mock and negative control (scrambled pre-miR) transfections were carried out for all experiments when cells were at 40–50% confluency. Scrambled pre-miR (AM17110) negative control and miR-1 (AM17150) precursor were purchased from Ambion (Life Technologies, Grand Island, NY).

Western blot analysis

Protein from whole cell extracts was extracted 24, 48, 72 and 96 h post-transfection via solubilization of A5 and B9 cells using RIPA buffer (50 mM Tris base pH 8, 150 mM NaCl, 1% NP40, 0.10% SDS) containing a protease inhibitor (Roche). For each sample, 30 µg of protein were run on a 10% polyacrylamide gel in a Bio-Rad mini-gel system at 150 V (12 V/cm) for ∼1.5 h. Proteins were transferred to nitrocellulose membrane for 70 min at 90 V (7 V/cm). Following transfer, membranes were blocked in 5% milk in TBST (10X TBS: 1.2% Tris, 8.8% NaCl) and were incubated with primary antibody overnight. Antibodies and dilutions were as follows: Ets1, 1:1000 dilution (antibody a gift from Michael Ostrowski) and α-tubulin, 1:100 dilution (T5168, Sigma-Aldrich, St. Louis, MO). Incubation with secondary antibodies was for 2 h. Secondary antibodies and dilutions were as follows: Anti-mouse HRP-Linked, 1:10000 dilution (W4021, Promega, Madison, WI), Anti-Rabbit HRP-Linked Antibody, 1:1000 dilution, (7074S, Cell Signaling Technology, Danvers, MA). Enhanced chemiluminescent reagent (Thermo Scientific, Rockford, IL) was applied in a 1:1 ratio.

Luciferase assays

Cloning. Cloning of Ets1 3’UTR was performed using In-Fusion Advantage PCR Cloning Kit according to manufacturer’s protocol (Clontech, Mountain View, CA). Full length Ets1 plasmid with 3’UTR provided by Dr. Michael Ostrowski was used as a template for PCR. A 1,278 base pair product was generated containing the predicted three miR-1 binding sites and was cloned into a pGL3 Control Luciferase vector (Promega, Madison, WI). Primers used for cloning were as follows: Ets1 3’ UTR In-Fusion Forward: 5’-tgtaatactagtccgAGAAGAGAGGCAATTGGCTGAGGT-3’, Ets1 3’ UTR In- Fusion Reverse: 5’-gtctgctcgaagcggACTGAGGCAGTATTCCTGATAGAG-3’. Clones were sequence verified. Site-directed mutagenesis was carried out to mutate two base pairs in each of the three miR-1 binding sites in the Ets1 3’UTR using QuikChange Lightning Multi Site-Directed Mutagenesis kit according to the manufacturer’s protocol (Agilent Technologies, Santa Clara, CA). Primers used for site-direct mutagenesis are as follows:

Ets1 3’ UTR SDM 1 Forward: 5’-gttgatggctgactcccactctcccttgaagactctgaat-3’,

Ets1 3’ UTR SDM-1 Reverse: 5’-attcagagtcttcaagggagagtgggagtcagccatcaac-3’,

Ets1 3’ UTR SDM-2 Forward: 5’-ccacaaggaagcaaaggccaaactctccagctatatattttgatctta-3’ ,

Ets1 3’ UTR SDM-2 Reverse: 5’-taagatcaaaatatatagctggagagtttggcctttgcttccttgtgg-3’,

Ets1 3’ UTR SDM-3 Forward: 5’-acctgttgaactcttacgtactctccaaagacgtttcaaggaac-3’,

Ets1 3’ UTR SDM-3 Reverse: 5’-gttccttgaaacgtctttggagagtacgtaagagttcaacaggt-3’.

Clones were sequenced verified.

Transfections and luciferase assays. C5N and A5 cells were plated in triplicate into a 12-well dish at 24 h prior to transfection. At 70% confluency, 0.10 µg Luc-Ets1 3’UTR and Luc-Ets1 3’UTR-mutated were each co-transfected with 10 pmol of a scrambled control miR or miR-1 into each well. 0.10 µg of a PRL-TK vector (TK-driven Renilla luciferase vector, Promega) was also transfected into each well to normalize the firefly luciferase values. All experiments included mock and pGL3-Control empty vector controls. A5 and C5N cell lysates were prepared using M-PER (Pierce Biotechnology, Rockford, IL) and 30 µg of each sample were used for analysis. Firefly and renilla luciferase measurements were acquired 24 h post-transfection using a Veristas Microplate Luminometer.

Cell proliferation assay

At 24 h post-transfection with miR-1, scrambled miR control or mock transfected, A5 and B9 cells were trypsinized and 2000 cells (A5) or 3000 cells (B9) were plated in quadruplicate in a 96-well plate. Proliferation was measured at 24, 48, 72 and 96 h post-transfection using a MTT (3-(4,5-Dimethylthiazol-2-yl)-2,5-diphenyltetrazolium bromide) cell proliferation kit (Roche, Indianapolis, IN). Experiments were carried out according to manufacturer’s protocol. RNA and protein were isolated at each solubilization step. Optimal absorbance of the formazan product was measured at a wavelength of 550 nm and a reference wavelength of 690 nm was used. For each sample, absorbance was normalized to the reference wavelength as well as a blank control, which contained media only.

Cell cycle assay

A5 and B9 cells were transfected with miR-1, scrambled miR control or mock transfected. Cells were trypsinized and counted 48, 72 and 96 h post-transfection. Two million cells were fixed in 70% ethanol. Cell cycle analysis was evaluated by DNA content. Cells were incubated with a propidium iodide solution (0.1% Triton X-100, 0.2 mg/mL DNAse-free RNAse A and 20 µg/mL propidium iodide) for 15 min at 37 °C. Data was acquired on a BD FACS Calibur instrument. ModFit software was used for analysis. Experiments were done in triplicate.

Apoptosis assay

A5 and B9 cells transfected with miR-1 or scrambled miR control were trypsinized and counted at 48, 72 and 96 h post-transfection. The FITC Annexin V Apoptosis Detection Kit I was used to evaluate cell death according to manufacturer’s protocol (BD Biosciences, San Jose, CA). Unstained, FITC Annexin only, and propidium iodide only conditions were used as controls. Data was acquired on a BD FACS Calibur instrument. Cell Quest Pro for was used for data analysis. Experiments were done in triplicate.

Cell motility assay

A5 and B9 cells were plated at a density of 7 × 105 cells/mL in ibidi cell culture inserts according to the manufacturer’s protocol (Martinsried, Germany) at approximately 48, 72 and 96 h post-transfection. Transfections of A5 and B9 cells included mock transfection, scrambled miR control (AM17110) or miR-1 precursor (AM17150) as described above. Eight hours after plating, cell inserts were removed. Images were taken at 5X magnification using AxioVision 4.8 software on an Axiovert 25 microscope at 0 h and every 5–8 h until gaps closed. Images were assessed visually by two independent researchers for differences in rate of gap closure.

Results

miRNA profiling of normal skin of cancer resistant and cancer susceptible mice

We hypothesized that miRNAs differentially expressed between cancer resistant and cancer susceptible mice might have oncogenic- or tumor suppressor-like properties in the skin. To identify miRNAs that were differentially expressed between cancer resistant SPRET/EiJ and cancer susceptible FVB/NJ mice, we performed miRNA expression profiling of total RNA isolated from four normal skin samples per strain. The miRNA arrays contained probes for precursor and mature miRNAs of approximately 300 miRNAs (Table S1; Liu et al., 2004; Liu et al., 2008). Following analysis, six miRNAs (miR-1, miR-133a, miR-124a-3, miR-134, miR-206, and miR-9-1) showed a significant difference in average expression and had a 2.0 fold or greater difference across one or more probe sets. All of these showed higher expression in SPRET/EiJ (Table 1). Due to the evolutionary divergence between M. musculus, for whom the majority of arrays have been designed, and M. spretus, lower miRNA expression observed in SPRET/EiJ mice could indicate a sequence polymorphism. However, we observed the opposite effect in which all miRNAs showing differential expression were higher in SPRET/EiJ. These results decrease the likelihood that the observed profiles were due to polymorphic variants between the strains.

Table 1 miRNAs showing statistically significant differential expression by array.

miRNA	Average fold difference
SPRET/FVB (range)	Types of probe	Speciesb	a Probes	p-valuec	
miR-1	2.6 (2.52–2.84)	Both	Both	8	1.8 × 10−5	
miR-124a-3	3.3	Mature	Mouse	1	1.9 × 10−7	
miR-133a	2.8 (2.71–2.92)	Both	Both	6	5.2 × 10−6	
miR-134	2.3	Prec	Human	1	2.4 × 10−6	
miR-206	2.6	Prec	Human	1	3 × 10−7	
miR-9-1	2.1	Prec	Mouse	1	1.7 × 10−5	
Notes.

a Average fold difference across all significant probe sets. Types of probes indicate whether probes for mature (mature), precursor (prec) or both (both) showed expression differences.

b Both mouse and human probe sets were included on the array. Species indicates whether probes for mouse and/or human showed expression differences.

c When multiple probes were significant, an average of the p-value for all the probes was calculated.

To validate the expression differences of the six miRNAs showing differential expression between SPRET/EiJ and FVB/NJ by microarray, we performed qPCR using Applied Biosystems TaqMan probes on samples from mice not used in the initial array studies. Probes were specific to either mature (miR-1, miR-133a, miR-124a-3, miR-206) or precursor (miR-9, miR-134, miR-206) miRNAs and were chosen for evaluation based on the specific probe sets (mature versus precursor) that showed differences by microarray (Table 1). Five of the six miRNAs evaluated showed similar fold-differences in expression between microarray and qPCR for FVB/NJ and SPRET/EiJ skin (Fig. 1 and data not shown); however, only miR-1, miR-133a and miR-206 showed statistically significant differences in expression by qPCR (p-value < 0.05).

Figure 1 Validation of candidate miRNA expression patterns in the skin by qPCR.

miRNA expression was measured by qPCR. miRNA expression in SPRET/EiJ skin (gray) relative to FVB/NJ skin (black) for five miRNAs are shown. Expression was normalized to sno-202 for the mature miRNAs and to L19 for the precursor RNAs. Fold expression was calculated relative to FVB/NJ. (A) miR-1, (B) miR-133a, (C) miR-124a, (D) miR-206 mature, (E) miR-206 pri, (F) miR-134 pri (M, mature, pri precursor). Error bars represent standard deviation. *, p-values for significance are shown for those microRNAs demonstrating p-values < 0.05 for differences in miRNA expression between FVB/NJ and SPRET/EiJ.

miRNA-1 expression decreased in mouse cSCCs

We hypothesized that miRNAs expressed at higher levels in SPRET/EiJ mice relative to FVB/NJ mice would behave similarly to tumor suppressor genes and exhibit reduced expression in tumors. We chose miR-1 for expression studies in mouse cSCCs to test our hypothesis because it showed a significant difference in expression between FVB/NJ and SPRET/EiJ for six probes on the array and showed a significant difference in expression by qPCR (Table 1). In addition, there was evidence from the literature showing down-regulation of miR-1 in a variety of human cancers and a link between miR-1 down-regulation and cancer phenotypes (Nasser et al., 2008; Datta et al., 2008; Yan et al., 2009; Nohata et al., 2012a; Nohata et al., 2012b). From these data, we expected that miR-1 expression would be decreased in mouse cSCCs. To evaluate miR-1 expression in cSCC, we performed qPCR of miR-1 in eleven cSCCs isolated from DMBA/TPA chemically-treated C57Bl6/FVB mice. The median miR-1 expression was lower in the tumors (3% of control sno-202) compared to median miR-1 expression in FVB/NJ (22% of control sno-202) and SPRET/EiJ (110% of control sno-202). The difference in expression was significant between the tumors and SPRET/EiJ (Fig. 2, p-value = 0.012). Based on the expression data in the primary tumors and the strain specific differences in expression, we chose miR-1 as a strong candidate miRNA to evaluate further for a role in cSCC.

Figure 2 Decreased miR-1 expression in cSCCs.

miR-1 expression was measured in eleven mouse cSCCs and six normal mouse skin RNA samples (3 FVB/NJ and 3 SPRET/EiJ) by qPCR relative to control gene sno-202. Median expression for each group is indicated by a line. p-value for significance between the tumors and SPRET/EIJ is indicated and was calculated using a nonparametric Mann-Whitney U test.

qPCR of miR-1 candidate target genes

miRNAs are thought to impact gene regulation by targeting mRNAs for degradation via complete sequence matches or inhibition of translation when there is imperfect binding between a miRNA and its target sequence (Djuranovic, Nahvi & Green, 2011). We hypothesized that if the difference in miRNA expression between the strains of mice was important in cancer susceptibility there would be differences in target mRNA expression of genes associated with carcinogenesis. To identify putative targets of miR-1, miR-133a, miR-124a-3, miR-134, and miR-206, we searched the literature to identify targets of these miRNAs that had been detected by expression arrays in tumors and had been further validated by other methods. We also identified genes predicted to be targets in microRNA databases www.microRNA.org (Beta et al., 2008) and www.targetscan.org. We prioritized genes as candidates which had two or more predicted binding sites, had stronger binding scores and those genes previously implicated in cancer. Using these strategies, we identified multiple candidate targets for each miRNA (Table S2).

We chose seven putative miR-1 targets, Bag4, Ets1, Met, Sp1, Taok1, Trp53, and Zfp148, and one positive control, Twf1, which previously had been shown to be a miR-1 target by Applied Biosystems, for testing. These candidate target genes were prioritized based on our database searches and those which had been validated in the literature as being targets of miR-1 in studies of primary human tumors or cancer cell lines. By qPCR, two of the candidate targets, Ets1 and Met, and our positive control Twf1 showed significant down-regulation in miR-1 transfected A5 cells compared to scrambled miR precursor transfected cells at both 48 and 72 h post-transfection (Fig. 3 and Fig. S1, Met p-values 0.0001 and 0.002; Ets1 p-value = 0.0001 and 0.0002; Twf1 p-values = 0.0001 and 0.006 respectively). Our positive control, Twf1, showed decreased expression in miR-1 transfected B9 cells at 48, 72 and 96 h (p-values = 0.0004, 0.0002 and 0.03 respectively). In B9 miR-1 transfected cells Met showed significantly decreased expression only at 48 h (p-value = 0.001), and Ets1 showed significantly decreased expression at 48 and 72 h post-transfection compared to scramble miR control transfected cells (p-values 0.008 and 0.005 respectively). Bag4 mRNA expression was significantly reduced in B9 cells transfected with miR-1 at all three time points (Fig. S1; 48 h p-value = 0.0007, 72 h p-value = 0.02 and 96 h p-value = 0.004), but it only showed significant down-regulation of mRNA at 72 h in A5 cells transfected with miR-1 compared to scrambled control cells (Fig. 3D and Fig. S1, p-value = 0.01). The remaining targets did not show statistically significant differences in expression between control and miR-1 A5 transfected cells (data not shown).

Figure 3 Validation of candidate miR-1 target genes by qPCR.

Fold mRNA expression of predicted targets of miR-1 (A) Met, (B) Ets1, (C) Twf1 and (D) Bag4 were assessed by qPCR. Expression is plotted as a fold difference in expression compared to the 48 h SC transfected cells. Expression was normalized to control gene L19. A5 cells transfected with a scrambled control miR (SC) are shown in black and A5 cells transfected with miR-1 are shown in gray. Cells were harvested at 48 and 72 h post-transfection. Difference in expression was measured by student’s T-test. Error bars represent standard deviation. *, p-value < 0.01, **, p-value < 0.001.

Evaluation of Ets1 as a miR-1 target

mRNA expression data is only correlative and does not show a direct effect of miRNAs on predicted target gene expression. As Ets1 had not been confirmed as a target of miR-1 in the mouse, and since several studies support a role of Ets1 in cSCC and cancer phenotypes such as metastasis, apoptosis and proliferation, we decided to further evaluate murine Ets1 as a miR-1 target (Keehn, Smoller & Morgan, 2004; Hahne et al., 2009; Nagarajan et al., 2009). We first assessed whether Ets1 protein levels were decreased by miR-1 expression. By Western, Ets1 was down-regulated in the miR-1 transfected A5 cells at 24, 48 and 72 h post-transfection and in B9 cells at 48 and 96 h post-transfection compared to scramble control miR transfected cells (Figs. 4A and 4B; Fig. S2). As this could be an indirect effect of miR-1 expression, we assessed the effect of miR-1 expression on a luciferase construct containing the 3’UTR of Ets1. miR-1 expression resulted in lower luciferase expression with the wildtype Ets1 3’UTR compared to a scrambled precursor miRNA. Furthermore a construct containing mutated miR-1 binding sites did not show reduced luciferase expression providing additional evidence that miR-1 regulates Ets1 expression via direct binding to the predicted miR-1 binding sites in the Ets1 3’UTR (Fig. 4C). Our data is consistent with a publication showing Ets1 to be a target of miR-1 in a human liver cancer cell line (Wei et al., 2012).

Figure 4 Evaluation of Ets1 as a miR-1 target.

Western blot analysis of Ets1 expression (A) 72 h after transient transfection of A5 cells with scrambled control miR (SC) or miR-1 and (B) 48 h after transient transfection of B9 cells with (SC) or miR-1. alpha-tubulin (A5) or Gapdh (B9) was used as a loading control. (C) Wild type (Ets1 3’UTR) and mutated (Ets1 3’UTR SDM) Ets1 3’UTR luciferase constructs were co-transfected with scrambled control miR (SC) or miR-1 into C5N cells. Data is expressed as relative light units of firefly over renilla luciferase. (C) The mouse Ets1 3’UTR has three miR-1 binding sites. Shown are the two base pairs (bold/underlined) in the seed region of each site that were mutated using site-directed mutagenesis. Error bars represent standard deviation. **, p-value < 0.01.

Functional characterization of miR-1 expression

miR-1 expression in cSCC cells decreases cell proliferation

There are several published studies showing in vitro and in vivo effects of miR-1 on tumor suppression (Nasser et al., 2008; Datta et al., 2008; Yan et al., 2009; Nohata et al., 2011; Nohata et al., 2012a; Nohata et al., 2012b). Based on these, we anticipated that expression of miR-1 in cSCC cells would have tumor suppressing abilities (Nohata et al., 2012b). To determine if miR-1 decreased cell proliferation, we transfected A5 and B9 cSCC cells with miR-1 or scrambled precursor miRNA. Cell proliferation was measured by an MTT assay at 24, 48, 72 and 96 h post-transfection. We observed a significant decrease of 1.7-fold in absorbance at 96 h post-transfection in A5 miR-1 transfected cells compared to the scrambled control miR transfected cells suggesting that miR-1 is playing a role in growth inhibition (Fig. 5A, 72 h p-value = 0.01; 96 h p-value = 0.0001). A modest, but significant decrease in proliferation was observed at the 72 h time point in the B9 cSCC cell line, but there was no difference in proliferation at 96 h (Fig. 5B, 72 h p-value = 0.02; 96 h p-value = 0.3).

Figure 5 miR-1 reduces cell proliferation in vitro.

An MTT cell proliferation assay was conducted using precursor miR scramble (SC) and miR-1 transfected in (A) A5 cells and (B) B9 cells. Hours post-transfection are indicated. Relative absorbance was measured at the indicated time points. We observed significantly decreased cell proliferation in the A5 miR-1 transfected cells at 96 h post-transfection (p-value, < 0.0001) and for the B9 miR-1 transfected cells at 72 h post-transfection (p-value < 0.01). Error bars represent standard deviation. Black diamond, scrambled control miR transfected; gray square, miR-1 transfected; *, p-value < 0.01; ***, p-value < 0.0001.

The effect of miR-1 expression on additional tumor phenotypes

miR-1 has been shown to induce apoptosis in multiple cancer cell lines including maxillary sinus SCC, head and neck SCC, and renal cell carcinoma (Nohata et al., 2011; Kawakami et al., 2012; Nohata et al., 2012a; Nohata et al., 2012b). To determine if the decreased proliferation seen in the cells transfected with miR-1 was the result of increased apoptosis, we performed FACS analysis of AnnexinV and PI stained miR-1 transfected and scramble control miR transfected A5 and B9 cells. No significant differences in apoptosis were observed at 48 or 72 h post-transfection in the A5 cells; however, there was a significant increase in apoptosis at 96 h (p-value = 0.036; Fig. 6A and Fig. S3). The B9 cells showed a statistically significant decrease in apoptosis in the miR-1 transfected cells at 48 h (p-value = 0.04) and a statistically significant increase in apoptosis at 72 h (p-value = 0.003). There were no differences in apoptosis at 96 h (Fig. 6B and Fig. S3). The apoptosis data for the A5 and B9 cell lines are consistent with the proliferation results.

Figure 6 Effect of miR-1 expression on tumor phenotypes.

Apoptosis in miR-1 or scrambled control miR transfected cells (SC) was measured at 48, 72 and 96 h by FACS analysis in (A) A5 and (B) B9 cells. The percentage of apoptotic cells staining positive for AnnexinV and propidium iodide is indicated in scrambled control (SC) (black) and miR-1 transfected cells (gray) for representative experiments is shown. Cell cycle parameters for SC (black) and miR-1 transfected A5 cells (gray) were measured in A5 cells at (C) 48 h, (E), 72 h and (G) 96 h and in B9 cells at (D) 48 h, (F) 72 h, and (H) 96 h by staining with propidium iodide and sorting via flow cytometry. The percentage of gated cells for G0-G1, S and G2-M phases for representative experiments are indicated. There was a statistically significant difference for S-phase for the A5 cells at all three time points. Error bars represent standard deviation. *, p-value < 0.05; **, p-value < 0.001; NS, not significant.

Ectopic expression of miR-1 has also been shown to induce G0/G1 arrest in several different cancer cell lines (Nohata et al., 2012b). We evaluated cell cycle parameters in miR-1 transfected cells by FACS analysis (Fig. 6 and Fig. S4). We observed modest but consistent and statistically significant fewer cells in the S phase for miR-1 transfected A5 cells compared to scrambled control miR transfected A5 cells at 48, 72, and 96 h (p-values = 0.002, 0.0001 and 0.006 respectively), Figs. 6C, 6E and 6G and for miR-1 transfected B9 cells at 48 and 96 h (p-values = 0.001 and 0.007 respectively), Figs. 6D and 6H. There was a trend for a modest increase in cells in G0/G1 for miR-1 transfected A5 cells at all time points, but this was only significant at 48 h post-transfection (p-value = 0.04); Fig. 6C. This was not observed in the B9 cells. There was a trend for more cells in G2-M in the miR-1 transfected cells for both A5 and B9 for some time points (Fig. 6 and Fig. S4), but the absolute differences are quite small and were not significant except for the miR-1 transfected B9 cells at 48 h (p-value = 0.02), Fig. 6F. These results suggest that miR-1 may have an extremely modest effect on cell cycle progression resulting in fewer cells in S phase.

Previous studies have reported a role of miR-1 in migration. In addition, Ets1 has a well-documented role in enhancement of migration (Hahne et al., 2005; Smith et al., 2012). To test whether miR-1 affects migration in cSCC, we performed cell motility assays of miR-1 transfected and scrambled control transfected A5 and B9 cells at approximately 24, 48, 72 and 96 h post-transfection. No consistent differences were observed in cell motility between miR-1 and scrambled control miR transfected cells for any of the time points (Fig. 7 and Fig. S5 and data not shown). Closure of gap appeared to be directly proportional to gap width.

Figure 7 Effect of miR-1 on cell motility.

Cell motility for miR-1 and scrambled control (SC) transfected A5 cells are shown for indicated time points after removal of insert at approximately (A) 48 (B) 72 h or (C) 96 h after transfection. Representative experiments for each time point are shown.

miR-1 expression following transfection was confirmed for all phenotypic experiments by qPCR and is below detectable limits for the mock or scrambled control transfected A5 and B9 cell lines.

Discussion

Exploitation of genetic differences between cancer resistant and cancer susceptible mice has led to the identification of candidate genes important in tumorigenesis, but little has been explored regarding the differences in miRNA expression profiles in these mice. Here, we describe our discovery of five miRNAs showing higher expression in skin cancer resistant mice compared to skin cancer susceptible mice. In more detailed analyses of one miRNA, miR-1, we show that miR-1 expression leads to decreased cell proliferation which may be the result of modestly higher apoptosis in later time points (72 or 96 h) in combination with a reduced number of cells in S phase. We also observed decreased mRNA expression of previously described human miR-1 targets, Met, Twf1, and Ets1, in the mouse which establishes that miR-1 targets these mRNAs in more than one species. Furthermore, we show that Bag4, a novel miR-1 target, is significantly down-regulated in the B9 cell line.

miR-1, miR-133a, and miR-206 are all part of a group of myo-miRs, miRNAs whose expression is enriched in skeletal and cardiac muscle (McCarthy, 2008). All three of these miRNAs showed higher expression in cancer resistant SPRET/EiJ compared to FVB/NJ normal skin. Initial studies on miR-1 focused on its expression in heart and skeletal muscle as it was thought not to be expressed in other tissues. In cardiomyocytes miR-1 regulates apoptosis via Bcl-2 and IGF-1 (Yu et al., 2008; Tang et al., 2009). In skeletal muscle, miR-1 has a role in cellular proliferation and differentiation (Chen et al., 2006). All of these functions are also important in tumorigenesis and are consistent with a role for miR-1 in cancer. A connection between expression of myo-miRs and skin cancer susceptibility has previously been made; a gene network for skin cancer susceptibility from the same strains of mice studied here was found to be enriched in muscle related mRNAs (Quigley et al., 2009).

Decreased miR-1 expression has been observed in a number of cancers including skin, lung, liver, bladder, renal and prostate (Li et al., 2012; Hudson et al., 2012; Nohata et al., 2012a; Tominaga et al., 2013; Nohata et al., 2012b). This is not the first study to look at the role of miR-1 in the skin or in SCCs. By array analysis, miR-1 showed about a 2-fold down regulation in human cSCCs compared to matched normal skin (Darido et al., 2011). miR-1, along with miR-133a, miR-205 and let-7d, showed decreased expression in SCCs of the head and neck in comparison to normal adjacent tissue (Childs et al., 2009). Consistent with their down-regulation in tumors, numerous studies have demonstrated tumor suppressor functions of miR-1. Increased expression of miR-1 in vitro has been associated with increased apoptosis, decreased migration and decreased cell growth (Hudson et al., 2012; Wei et al., 2012; Yamasaki et al., 2012; Yoshino et al., 2012). miR-1 was also found to target c-Met in rhabdomyosarcomas and colorectal cancer leading to decreased cell motility and proliferation (Yan et al., 2009; Reid et al., 2012). Our results showed decreased Met mRNA expression in cells transfected with miR-1, but observed no correlation between miR-1 expression and cell motility. In this study, we identified a correlation with proliferation and miR-1 expression levels which is consistent with other studies (Yan et al., 2009; Reid et al., 2012).

In contrast to previous studies we observed only modest effects of miR-1 on apoptosis and cell cycle and did not observe any consistent effects on migration. We observed a consistent decrease in the percentage of miR-1 transfected cells in S-phase and a non-significant increase of miR-1 transfected cells at G2/M compared to scrambled control cells for some time points, but these differences were very modest compared to studies by others (Nohata et al., 2011; Hudson et al., 2012; Kawakami et al., 2012; Wei et al., 2012). We also observed increases in apoptosis in both B9 and A5 cells transfected with miR-1, but the apoptosis was only observed at later time points and the percentage of cells undergoing apoptosis was less compared to other studies (Hudson et al., 2012; Wei et al., 2012; Yoshino et al., 2012). Finally, we observed no significant differences in cell motility of miR-1 transfected A5 or B9 cells. There are several potential explanations for the observed results. First, these cell lines may not be as responsive to the effects of miR-1 because they lack high expression levels of targets. Some miR-1 oncogene-like targets have been identified that were not evaluated for expression in our cells. However, many of the targets tested in our study that have been reported to have oncogenic properties, such as Bag4, Ets1 and Met, showed decreased expression in A5 and/or B9 cells following miR-1 transfection. Another possible explanation for the discordant results is that we evaluated the effects of miR-1 expression in mouse and not human cell lines. Most of the published studies on miR-1 have been in human cancers and using human cell lines and it is possible that miR-1 has unique targets in the mouse which confer tumor suppressor activities that ameliorate some of the effects of silencing target oncogenes. Additional studies are warranted to further explore these and other possibilities.

Ets1 is a transcription factor that has been described as a proto-oncogene due to its pro-tumorigenic effects and its discovery in chickens affected with the avian erythroblastosis virus, E26 (Dittmer, 2003). Specifically, Ets1 has been shown to play a role in cSCC tumor development and progression (Pande et al., 1999; Nagarajan et al., 2009). Ets1 acts as a cSCC tumor promoting gene by inducing cSCC cell proliferation in transgenic mice over-expressing Ets1 (Pande et al., 1999). Studies show that Ets1 regulates genes important in apoptosis, angiogenesis, migration, invasion, and matrix metalloproteinases which enhance cell migration (Pande et al., 1999; Keehn, Smoller & Morgan, 2004; Hahne et al., 2005). In addition, it is over-expressed in human cSCC malignancies (Keehn, Smoller & Morgan, 2004; Saeki et al., 2000; Saeki et al., 2002). A previous study looked at Ets1 expression in 26 primary skin lesions. Ets1 nuclear expression was present in cSCC in situ, but was the highest in poorly differentiated or metatstatic cSCCs (Keehn, Smoller & Morgan, 2004). These results suggest the importance of dysregulation of Ets1 expression for progression of cSCC. Consistent with a role for miR-1 in dysregulation of Ets1 in cancer, a study published after we chose Ets1 as a candidate showed that Ets1 is targeted by miR-1 in HepG2 cells (Wei et al., 2012). Ets1 is involved in regulation of numerous protein tyrosine kinases (PTKs), a class of molecules to which a majority of known oncogenes belong (Hahne et al., 2009). Some predicted Ets1 targets include EGFR, FGFR4, INSR, MET, PDGFRA, and VEGFR1 (Hahne et al., 2009). Ets1 has been shown to be involved in regulation of tumorigenesis by promotion of the EGFR-Ras-MEK1/2 MAPK pathway (Roberts & Der, 2007; Ciardiello & Tortora, 2008; Montagut & Settleman, 2009; Knight, Lin & Shokat, 2010). Our positive control for miR-1 transfection, Twf1, also belongs to the family of PTKs regulated by Ets1. As miR-1 has multiple experimentally validated targets from this study (Ets1, Met, Bag4) and from published studies (MET, PTK1, HDAC4, ANXA2, BDNF and FOXP1) that have roles in tumorigenesis, it is likely that miR-1’s ability to reduce the expression of many tumor promoting genes could have a global influence on the suppression of tumor development (Nasser et al., 2008; Datta et al., 2008; Yan et al., 2009; Reid et al., 2012).

Our data suggests that Bag4 may also be a target of miR-1 which is a novel finding. Like Ets1 and Met, Bag4 has been shown to have transforming properties (Yang et al., 2010). Bag4 is a member of a family of proteins that act as co-chaperones for anti-apoptotic proteins Bcl-2 and Hsp70 (Doong, Vralias & Kohn, 2002). The literature describing the role of BAG4 in apoptosis is mixed as BAG4 is associated with apoptosis in some studies, but appears to be anti-apoptotic in others (Ozawa et al., 2000; Annunziata et al., 2007). Additional studies are warranted to further evaluate Bag4 as a miR-1 target and any role it may have in cSCC.

There are some limitations to this study. Only one of the three significantly differentially expressed miRNAs by qPCR was the subject of further phenotypic studies. miR-133a and miR-206 may play important roles individually or in combination with miR-1 in the development of cSCC which were not evaluated in this study. Another weakness of this study was that candidate targets of miR-1 were chosen based on the literature and predicted in silico screens rather than experimentally. Thus, we may have missed critical tumor suppressing targets of miR-1 through this approach.

We identified and validated five miRNAs showing differential expression between cancer resistant and cancer susceptible mice. The mice used in the study are inbred, are age and sex-matched, and should show very similar expression profiles between mice within a strain, but it is possible that the inclusion of additional mice in the study would have led to increased power and the identification of additional miRNAs of interest. As we used a 2-fold difference in expression for our cutoff we would miss miRNAs that showed more modest differences in expression between Spret/EiJ and FVB/NJ. Nonetheless, these data are consistent with a study measuring miRNA expression of 577 miRNAs from livers of 70 different strains of inbred mice which found few to no differences in expression of any of these miRNAs (Gatti et al., 2011). A few studies have looked at miRNA expression in human cSCCs, but beyond miR-1 none of the miRNAs identified in those studies overlap with the ones identified here (Dziunycz et al., 2010).

It is interesting to note that in our original analysis, we identified miRNAs for which the precursor, but not the mature form of the miRNA showed differential expression on the array. This discrepancy may reflect transcription differences between the strains followed by miRNA processing resulting in similar amounts of mature miRNA being produced, or the precursor miRNAs may also be processed into different forms of mature miRNAs, such as 5p and 3p forms, for which there were not probes on the array. These data may be a reflection of our study being statistically underpowered to detect some of the differences in miRNA expression. For example, miR-206 showed statistical differences in expression of the precursor miRNA but not the mature miRNA by microarray, but by qPCR both the precursor and mature miR-206 showed very similar differences in expression between FVB/NJ and SPRET/EiJ.

In summary, we identified five miRNAs that show differences in expression of skin between cancer resistant and cancer susceptible mice. We show evidence that miR-1 expression is decreased in cSCCs and that its expression in vitro leads to decreased proliferation of cSCC cell lines, increased apoptosis and modest effects on cell cycle. In cell lines, miR-1 expression is correlated with decreased Met1 and Bag4 expression and directly targets the oncogene Ets1. This is the first study to show that mouse Ets1, like the human ETS1 gene, is targeted by miR-1. Our studies showing similar miR-1 targets between the mouse and human suggest that mouse models of miR-1 may be useful in elucidating the role of miR-1 in cancer. As the endogenous expression of miR-1 differs between skin cancer susceptible and cancer resistant mice, it may play a role in the differences in cancer risk observed in these strains, but the exact mechanism remains to be elucidated.

Supplemental Information

Figure S1 Candidate miR-1 target genes in B9.

mRNA expression of predicted targets of miR-1 (A) Met, (B) Ets1, (C) Twf1 and (D) Bag4 were assessed by qPCR. B9 cells transfected with a scrambled control miR (SC) are shown in black and B9 cells transfected with miR-1 are shown in gray. Cells were harvested at 48, 72 and 96 h post-transfection. Expression is plotted as a fold difference in expression compared to the 48 h SC transfected cells. Expression was normalized to control gene L19. Difference in expression was measured by student’s T-test. Error bars represent standard deviation. *, p-value < 0.01, **, p-value < 0.001, NS, not significant

Click here for additional data file.

Figure S2 Ets1 levels following miR-1 transfection b.

(A) Western blot analysis of Ets1 expression 24, 48 and 72 h after transient transfection of A5 with scrambled control miR (SC) or miR-1. (B) Western blot analysis of Ets1 expression 24, 48 or 96 h after transient transfection of B9 with SC or miR-1. Gapdh was used as a loading control.

Click here for additional data file.

Figure S3 AnnexinV analysis of apoptosis.

Apoptosis in miR-1 or scrambled control miR transfected cells (SC) was measured at 48, 72 and 96 h by AnnexinV and PI staining measured by FACS analysis. Shown is a representative sample at 72 h post-transfection for the A5 (A) SC and (B) miR-1 transfected cells and for the B9 (C) SC and (D) miR-1 transfected cells. The gating of the cells and the percentage of apoptotic cells staining positive for AnnexinV, propridium iodine or both (upper right quadrant) are shown.

Click here for additional data file.

Figure S4 FACS analysis of cell cycle.

Cell cycle parameters for scrambled control miRNA (SC) and miR-1 transfected cells were measured for A5 cells at (A) 48 h, (B), 72 h and (C) 96 h and for B9 cells at (D) 48 h, (E) 72 h, and (F) 96 h by staining with propidium iodide and sorting via flow cytometry. The percentage of gated cells for G0-G1, S and G2-M phases are indicated.

Click here for additional data file.

Figure S5 Effects of miR-1 on B9 cell motility.

Cell motility for miR-1 and SC transfected B9 cells are shown for 0, 8, 11, and 13 h after removal of insert at approximately (A) 48 h (B) 72 h, (C) or 96 h after transfection. Representative experiments for each time point are shown.

Click here for additional data file.

Table S1 miRNA expression array data.

miRNA array data for three FVB/NJ, three SPRET/EiJ samples and one replicate RNA sample (FVB-A or –B and SPRET-A or B) per strain.

Click here for additional data file.

Table S2 Predicted target mRNAs.

Genes predicted to be targets of the validated five miRNAs.

Click here for additional data file.

We thank John Hagan for his assistance in analysis of the microarray data and Hee-Yeon Cho for help in validation of microRNA expression profiles. Dr. Michael Ostrowski provided the Ets1 expression construct and the Ets1 antibody. Dr. Allan Balmain provided the A5, B9 and C5N cell lines and mouse cSCC RNA. The OSU Comprehensive Microarray Shared Resource performed the miRNA expression arrays, FACS analysis was performed in the OSU Flow Cytometry Shared Resource, and the qPCR plates were run in the OSU Comprehensive Cancer Center Nucleic Acids Shared Resource.

Additional Information and Declarations

Competing Interests

Author Contributions

Animal Ethics

Microarray Data Deposition

Dr. Toland is an Academic Editor for PeerJ.

Jessica L. Fleming and Dustin L. Gable conceived and designed the experiments, performed the experiments, analyzed the data, wrote the paper.

Somayeh Samadzadeh-Tarighat conceived and designed the experiments, performed the experiments, analyzed the data.

Luke Cheng and Jessica L. Gillespie performed the experiments, analyzed the data.

Lianbo Yu conceived and designed the experiments, performed the experiments, analyzed the data, contributed reagents/materials/analysis tools.

Amanda Ewart Toland conceived and designed the experiments, analyzed the data, wrote the paper.

The following information was supplied relating to ethical approvals (i.e. approving body and any reference numbers):

Pre-existing or commercially available animal specimens were used for this study. These studies were approved by the Ohio State University (The Ohio State University IACUC has determined that your research is exempt from IACUC review) and University of California, San Francisco Institutional Animal Care and Use Committees (IACUC approval number AN084982).

The following information was supplied regarding the deposition of microarray data:

Data is attached as Supplemental Information.

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
