# Peer review of "Differential expression of miR-1, a putative tumor suppressing microRNA, in cancer resistant and cancer susceptible mice"

_PeerJ, doi:10.7717/peerj.68_

## Round 0.1 · original submission · Major Revisions

All three reviewers had major concerns about the quality of the manuscript; one actually suggested to reject the paper. I am aware of the fact that PeerJ is a new journal and that the paper had to be prepared on short notice. Also, PeerJ has a policy with respect to publication that is different to most other journals. Still, the criticism of the reviewers was so substantial, and justified, that a real major revision is required to make the manuscript suitable for publication.

Reviewer 1 ·

Basic reporting

In this article, the authors examined the differential miRNA expression between Mus Musculus and Mus Spretus and identified multiple miRNAs that have significantly different expressions, including miR-1. They further tested miR-1’s function on predicted target genes and identified Ets1 as a valid target gene. The author also analyzed miR-1’s function in cSCC cells with proliferation assays, cell cycle assays and wound healing assays.
Below are the issues the authors should address:
1. In Fig 1, the authors claim the expressions of all six miRNAs have significant difference. However, the qPCR results only show 3 miRNAs with p-value less than 0.05. The authors could repeat the experiments a few more times and get significant p-values.
2. The authors failed to define the designation of the error bars in any figures. Do they indicate SD, SEM or something else?
3. There is no error bars in Fig 2. Is each result based on single experiment? FVB/NJ and SPRET/Eij each has two sets of normal skin data. Why do the data from the same sample vary so much? The authors should do more repeats and show one bar for each sample with error bars, and calculate the p-value between samples or groups they want to compare.
4. In Fig 4, the authors mutated all three potential miR-1 sites in Ets1 3’UTR and showed no inhibition in the mutated 3’UTR. The issue here is that not all predict target sites are effective. To determine which one or ones are true sites, the authors should show the results of luciferase assays with 3’UTRs that contain each one mutation and then the double mutation.
5. In Fig 6, although the authors claim consistent increase of G2-M cells and decrease of G0-G1 cells in miR-1 transfected samples, the p-values are not given in 6C and 6D. 6D even lacks error bar. Is it just a single experiment? The authors should also show how they gated the cell in a bi-variate plot, for many people use wrong gates in their analyses.
6. In Fig 5 and 6, the growth curves are at odds with the apoptosis and cell cycle assays. B5 cells’ growth is not affected but there is significant change of apoptosis, and the authors claim G1 arrest. How do the author explain the contradictory results?
7. In Fig 6E, which cell is used in the wound healing assay, A5 or B9? The authors should show both results anyway.
8. Western blot and qPCR confirmation of Est knockdown in A5 and B9 cells are required to demonstrate the effectiveness of the transfection for functional assays. What cell lines did the authors use for the Western blot and qPCR in Fig 4? If it is A5 or B9, the authors should show a set of data in the other cell line.

Experimental design

see above

Validity of the findings

see above

Additional comments

see above

Reviewer 2 ·

Basic reporting

The Paper "Differential expression of miR-1, a putative tumor suppressing microRNA, in cancer resistant and cancer susceptible mice" is an effort to establish miR-1 as a putative tumor suppressor microRNA in a skin cancer model of cancer resistant and cancer susceptible mice.

Experimental design

The study lacks an unbiased approach. The authors are not convincing why they only selected miR-1 to study forward out of 3 statistically validated miRs. Then why only Ets-1 was selected for further studies, when there are multiple putative targets. Simple approach could be to have overexpression (which they have) and check with microarray, there would be direct and indirect targets, but unbiased approach. The targets can be validated with the knockdown too.

Validity of the findings

Are the identified miRNAs can be mapped in QTLs?
Lack of statistical analysis, Biological replicates and Technical replicates.
The real-time PCR data represented as percentage Relative expression, I am really not comfortable with that. Fold Change; Relative mRNA expression would be the better way to represent. The qPCR for the miRs should be clarified which gene was used for the normalization, the house keeping gene selected is for the normalization (relative should be with the normal or the control sample).
Scatter plot would be better for the variations in the experiment. The experiments lack proper statistical analysis in many figures.
The overall study is based on maximum 2 fold differential expression of six miRs. That is less convincing for a phenotypic role.

Additional comments

The Paper lacks major merits to be published in the submitted form. Different fonts in the same paragraph. Material methods have lots of repetitive sentences. Overall write-up needs to be in proper tense, Just few examples, We incubated/We saw/gel transfer to NCP took place for. It’s a poor write up throughout in the Material and Methods, Results and the Figure legends. Why the Table 1 is after Figure 1.

Reviewer 3 ·

Basic reporting

Not all relevant literature is referenced (see comments).

Experimental design

the technical standards are not always high
not all methods have been described with enough detail to allow reproduction

Validity of the findings

most data it not significant, the selected time points complicate integration of individual data
the conclusions are not supported by the data. In fact, the authors put their whole study in question when they discuss the possible causes of their observations

Additional comments

Fleming et al, identified miR-1 as a differentially expressed miRNA in skin tissues from cancer resistant vs cancer susceptible mice and they validated MET and ETS1 as targets of miR-1 in their murine skin cancer cell line A5. Although these findings are in principle interesting, they are not novel, as previous studies has demonstrated that miR-1 is downregulated in skin tumors vs normal tissue (Darido et al, 2011), while MET and ETS1 are already known miR-1 targets (Yan et al, 2009; Wei et al, 2012). The authors focus on just one of their miRNA hits and then only on one of its targets. However, the functional tests do not provide convincing data to support all claims. Furthermore, no evidence is shown that Ets1 would indeed be related to the phenotypes that are analyzed. The authors remain at the miRNA-level without digging a little into the targets that could be suspected to indeed be relevant. Mir-1 has been shown to directly target the Ets1 transcription factor. There is some indication that this microRNA also regulates Met1 expression. Ets1 has previously been shown to drive expression of the Met1 gene. This triangle could be interesting to study in the context of feedback regulation. However, the cell line models that would be used in any potential further studies should be validated to be relevant for the disease system under investigation.
Major comments:
1. Why have the authors focused on already published targets of miR-1 and not on new ones? This should be discussed in the manuscript. A list of the predicted-promising candidate targets identified using miRANDA, expression profiling data (line 298) and that ‘had been further validated by other methods’ (lines 298/299) should be supplied as a table. The authors should also clarify which expression profiling arrays and other studies (references, DOI) they refer to. Line 302: these ‘multiple candidate targets’ should be provided. Line 321-322: what do the authors mean by saying that these targets were previously shown to be targets of miR-1 by Applied Biosystems (reference?!), for testing? Several of the targets are said to having been validated in previous studies. Instead of referring to supplementary table 1 (which I have been unsuccessful to find) the authors should provide the respective citations in the text. In general the description of the target selection is confusing and not clearly written.
2. Line 333: The authors claim that Ets1 has not yet been identified as a target of miR-1, however, a few lines below (line 346-347) they mention that it has indeed previously been shown to be a direct target of miR-1 in a human hepatocyte cell line (Wei et al., Oncol Rep. 2012 Aug;28(2):701-6).
3. The data obtained in this study do not support the claim of the authors that miR-1 may function as a tumor suppressor in (mouse) skin cancer. The authors show modest but not significant effects of miR-1 in various assays including proliferation, apoptosis, cell cycle analysis and migration. They observe a convincing effect on cell proliferation after overexpressing miR-1 in only one of the two cell lines used. In the discussion (lines 444-445) the authors hypothesize that the cell lines they picked as model systems might not be the right ones as these ‘do not express high levels of miR-1 targets or {would} have additional means to perturb those pathways’. Quantitative analysis of mRNA expression should have been performed for the putative target mRNAs to show that these genes are indeed expressed in the cell lines under investigation. At least Ets1 (and Met, Ptk9) seem to be expressed, otherwise the data shown in Figure 3 would not be possible. The authors should test more cell line systems and, preferentially, also a human one, to show that their results are also relevant in human skin cancer.
4. In line 452ff the authors speculate that miR-1 expression might be confined to non-tumor cells in complex tumor tissues (i.e., stroma) as a possible explanation for not having observed significant effects in their tumor cell line models. If this held true, the whole study they performed would not make sense at all. However, other studies have indeed shown differential expression of this miRNA in tumor cells (like the Wei paper cited above) and that this indeed has functional consequences. Further, they speculate that presence or absence of CpG islands would speak for (or against) regulation of miRNA expression by epigenetic mechanisms in man and mouse. Epigenetic regulation is not confined to CpG islands but rather mostly affects CpG dinucleotides that are outside such islands. Since the authors describe differences in the activities of miR-1 expression (or overexpression) in their mouse cell systems and not the regulation of miR-1 expression, the observed differences seen at the effector level should not be due to the endogenous regulation of that microRNA anyway. In lines 461ff the authors put their concept again in question. What sense would it make to publish data from a cell system when the relevance of this system in the biomedical context is not clear?
5. The results on cell-cycle analysis are not convincing. It is hard to believe that there is indeed a G2 arrest without proving statistical significance. In Figure 6D the error bars are missing (single experiment??). In lines 376/377 the authors state that ‘in some experimental replicates, statistically significant differences’ (were observed). I do not fully understand what the authors mean. The effects are either significant (that is why replicates are made and resulting data are analyzed in concert) or individual datapoints have been generated and compared (then, statistical analysis cannot be performed). If the authors have data that would support their claims they should show them instead of writing ‘data not shown’.
6. The fact that the authors do not observe tumor suppressive effects of miR-1 in these murine cell lines may well be context dependent. Some target-genes may not be conserved between man and mouse (Ets1 seems to be conserved). They also emphasize the role of Ets1 in skin cancer. It would be interesting to see if Ets1 inhibition by RNAi techniques, in murine skin cancer cell lines, has a tumor suppressive effect. For example, the authors should test the effects of knockdown of Ets1 in the cell migration assay. However, if they saw an effect of Ets1 there, this would not be regulated by miR-1…
7. The selection of time points in functional assays is not clear. Ets1 protein was found to be downregulated at 72 hours, unfortunately no data for the 48 hour timepoint are shown. The cell proliferation assay indicates first effects at 96 hours (in the text, first significant differences are said to having been observed at 72 hours, however, this is not supported by the data shown in Figure 5A – no indication of significance levels). Data shown for the B9 cell line are not convincing. The apoptosis assay was carried out at 48 hours post transfection. No clear indication of timing is given for the motility assay. Transfected cells are said to having been incubated over night before the scratch was set (removal of ibidi inserts). This would leave very little time to develop the phenotypic changes after knockdown of target genes. The endpoint of the wound-healing assay is obscure. The different timing of endpoints complicates integration and interpretation of individual experiments.
8. the authors hypothesize that elevated expression of miR-1 would lead to a G2-M arrest while previous studied have found this miRNA to rather affect G1-S transition. The authors should discuss this potential discrepancy, however, the cell cycle data they present is not really convincing.
Minor comments:
1. Text in lines 64-66 is somewhat redundant.
2. Lines 80-82. A reference for the OSUCCC version 4.0 microarrays should be given (Liu et al., 2004,2008 from line 265?). Why have only 300 human and mouse miRNAs been analyzed? What is the sense of testing for expression of human miRNAs in mouse tissues?
3. Lines 95f. why have miR-134, -206 and -9-1 not been quantified for mature forms?
4. Lines 170/171. Volts per cm should be given instead of total voltage applied during electrophoresis and electro-transfer of proteins.
5. Line 256: In the cell motility assay the cells were monitored over 24 hours with pictures being taken every 4-24 hours. There can be only one 24 hour time point. How has gap closure been quantified and how has the time of gap closure been determined? Commonly the endpoint is set at a timepoint prior to gap closure and the distance of migration is quantified.
6. Line 314: lower in the majority. The ‘in’ is missing.
7. Line 325: Twf1 showed significant down-regulation at the mRNA level in miR-1 transfected cells. Authors should use consistent naming of genes. Twf1 in the text is an alias of Ptk9 used in Figure 3. A5 cells were used to assess miR-1 effects on expression levels of putative target genes. What is the endogenous expression level of this microRNA in that cell line (and in the other cell lines used in this study)?
8. Line 372: I would not speak of an opposite effect when comparing to an insignificant observation.
9. Figure 3D: the error bar in Bag4 expression after miR-1 expression (72h) is huge. The authors should carry out more experiment to support their claim that Bag4 would be another target, data obtained at the 48h timepoint is not convincing along this line.
10. Line 495: The authors write that they found 300 miRNAs to have shown differential expression between cancer resistant and cancer susceptible mice. The grammar of that sentence is misleading as the array investigated only those 300 miRNAs and only five of these were indeed found to be differentially expressed.
11. In the figure 1, sub-figures G and H are missing from the figure although they are mentioned in the figure legend.
12. Line 351: The effect of miR-1 has been shown also in vivo and not only in vitro (e.g. in prostate cancer).
13. Lines 420 and 433: references are missing
14. Table 1. what is an ‘average p-value’?
15. Figure 3: the mRNA levels of the miR-1 targets should also be shown in B9 cells. Also, the labeling of the y axis is confusing. How was this %Relative expression calculated? Is it fold-changes that are shown? Which sample serves as a reference in order to calculate the fold change?
16. in the legend for Figure 4 the cell line tested for Ets1 expression after miR-1 transfections should be indicated
17. In Figure 6E the 0h timepoint should be shown

---

## Round 0.2 · accepted · Accept

Dear colleague,

The manuscript was re-reviewed by the reviewers, who had seen it before. You can see their comments below. Both suggested that the manuscript should be published in its revised form.

Reviewer 1 ·

Basic reporting

The paper is suitable to join the scientific literature. The experimental model is based on skin cancer model of cancer resistant and cancer susceptible mice.

Experimental design

See Below

Validity of the findings

See Below

Additional comments

The Paper "Differential expression of miR-1, a putative tumor suppressing microRNA, in cancer resistant and cancer susceptible mice" is an effort to establish miR-1 as a putative tumor suppressor microRNA in a skin cancer model of cancer resistant and cancer susceptible mice.

As mentioned by the author, the study was initiated in 2009, and the miRNA field has seen an exponential rise in reagents, methods and database analysis, the study is a reasonable effort to identify a putative tumor suppressor miRNA in a murine skin cancer model.

The authors have made sincere efforts and considerable changes to clarify the results and the comments from the reviewer. After going through the modifications, changes, addition of data points, explanations of the comments and supplementary data, considering the scope of the work, authors have answered all the questions raised.

The paper can be accepted in the submitted form.

Reviewer 2 ·

Basic reporting

The authors have addressed most of reviewers' comments. I thereby recommend acceptance.

Experimental design

na

Validity of the findings

na

Additional comments

na